# BEHIND THE MYTH OF EXPLORATION IN POLICY GRADIENTS

## ABSTRACT

Policy-gradient algorithms are effective reinforcement learning methods for solving control problems with continuous state and action spaces. To compute near-optimal policies, it is essential in practice to include exploration terms in the learning objective. Although the effectiveness of these terms is usually justified by an intrinsic need to explore environments, we propose a novel analysis and distinguish two different implications of these techniques. First, they make it possible to smooth the learning objective and to eliminate local optima while preserving the global maximum. Second, they modify the gradient estimates, increasing the probability that the stochastic parameter update eventually provides an optimal policy. In light of these effects, we discuss and illustrate empirically exploration strategies based on entropy bonuses, highlighting their limitations and opening avenues for future works in the design and analysis of such strategies.

## 1 INTRODUCTION

Many practical problems require making sequential decisions in environments, based on state observations, in order to minimize a cost or maximize a reward. Reinforcement learning is a framework for solving such decision-making problems that has been successful on complex tasks, including playing games (Mnih et al., 2015; Silver et al., 2017), controlling robots (Kalashnikov et al., 2018), or interacting with electricity markets (Boukas et al., 2021).

Reinforcement learning can be divided into three families of algorithms, namely, model-based, value-based, and policy-based methods. Each method exhibits different learning dynamics and requirements for computing high-performing policies. On the one hand, the first two families of algorithms are subject to the exploration-exploitation dilemma during the learning procedure. In short, in order to learn statistical estimates of the environment or the value functions as fast as possible, from which a good policy can be computed, it is necessary to take actions that increase the quality of the estimates that are likely not optimal. This need for exploration to achieve high performance is theoretically well understood and has been the subject of many works (Dann et al., 2017; Azar et al., 2017; Neu & Pike-Burke, 2020). On the other hand, in policy-based methods, and especially for policy-gradient algorithms (Duan et al., 2016; Andrychowicz et al., 2020), the main theoretical requirement to converge towards globally (or even locally) optimal solutions is that policies remain sufficiently stochastic during the learning procedure (Bhandari & Russo, 2019; Bhatt et al., 2019; Agarwal et al., 2020; Zhang et al., 2021a; Bedi et al., 2022). Interestingly, stochastic policies have smoother returns (Ahmed et al., 2019; Bolland et al., 2023), but neither softmax nor Gaussian policies guarantee enough stochasticity for ensuring (fast) convergence (Mei et al., 2020; 2021; Bedi et al., 2022). This requirement of stochasticity in policy gradient is often abusively called exploration and often understood as the need to infinitely sample all states and actions.

Practitioners have tried to meet the theoretical requirement of sufficient randomness of policies in policy gradient via reward-shaping strategies, whereby a learning objective that promotes or hinders behaviors by providing reward bonuses for some states and actions is optimized as a surrogate to the return of the policy. These bonuses typically promote actions that reduce the uncertainty of the agent about its environment (Pathak et al., 2017; Burda et al., 2018; Zhang et al., 2021c), or that maximize the entropy of states and/or actions (Bellemare et al., 2016; Lee et al., 2019; Guo et al., 2021; Williams & Peng, 1991; Haarnoja et al., 2019). Optimizing a surrogate objective is particularly effective for solving tasks with complex dynamics and reward functions, or with sparse

rewards (Islam et al., 2019; Lee et al., 2019; Liu & Abbeel, 2021; Zhang et al., 2021b; Guo et al., 2021).

The differences between theory and practical implementations of exploration has led to common folklore seeking to explain the intuition behind and the efficiency of policy gradient methods. This work is part of the research line that studies the maximization of practical surrogate learning objective functions from a mathematical optimization perspective. Close to our work, studies of the learning objective with entropy regularization (an exploration-based reward shaping technique where the entropy of the policy is added in the learning objective) were conducted. It includes the study by Ahmed et al. (2019) concluding that it helps to provide smooth learning objective functions. The same exploration strategy was reinterpreted as a robust optimization method by Husain et al. (2021) and equivalently as a two-player game by Brekelmans et al. (2022). Bolland et al. (2023) furthermore argued that optimizing an entropy regularized objective is equivalent to optimizing the return of another policy with larger variance. Chung et al. (2021) also studied the effect on the learning dynamics when including baselines in policy gradient, which is close to adding exploration terms in the learning objective. These studies are specific to some exploration methods and the literature lacks unified explanations and interpretations about exploration in policy gradient methods.

Before delving into our contributions, we recall that the convergence of stochastic ascent methods is driven by the objective function and how the ascent directions are estimated. First, the objective function shall be (pseudo) concave to find its global maximum (Bottou, 1998). Second, the convergence rate is influenced by the distribution of the stochastic ascent estimates (Chen & Luss, 2018; Ajalloeian & Stich, 2020). In this paper, we rigorously study policy-gradient methods with exploration-based reward shaping through the lens of these two optimization theory aspects. More precisely, we first discuss the effect of exploration on the learning objective and the relationship between an optimal policy and a policy maximizing the learning objective. Second, we elaborate on the distribution of the gradient estimates of the learning objective and its likelihood of providing a direction in which the learning objective and the return increase. We furthermore illustrate how some common exploration strategies help improve the performance of policy-gradient algorithms with respect to these two aspects. In practice, finding good exploration strategies is known to be problem specific and we thus introduce a general framework for the study and interpretation of exploration in policy gradient methods instead of trying to find the best exploration method for a given task.

The paper is organized as follows. In Section 2 we provide the background about policy gradients and about exploration. Section 3 focuses on the effect of exploration on the learning objective while Section 4 is dedicated to the effect on the gradient estimates used in the policy-gradient algorithms. Finally, conclusions and future works are discussed in Section 5.

## 2 BACKGROUND

In this section, we introduce the reinforcement learning problem in Markov decision processes and discuss the policy-gradient optimization method with exploration.

### 2.1 MARKOV DECISION PROCESSES

We study problems in which an agent makes sequential decisions in a stochastic environment in order to maximize an expected sum of rewards (Sutton & Barto, 2018). The environment is modeled with an infinite-time Markov Decision Process (MDP) composed of a state space $\mathcal{S}$, an action space $\mathcal{A}$, an initial state distribution with density $p_0$, a transition distribution (modeling the dynamics) with conditional density $p$, a bounded reward function $\rho$, and a discount factor $\gamma \in [0, 1($. When an agent interacts with the MDP, first, an initial state $s_0 \sim p_0(\cdot)$ is sampled, then, the agent provides at each time step $t$ an action $a_t \in \mathcal{A}$ leading to a new state $s_{t+1} \sim p(\cdot|s_t, a_t)$. Such a sequence of states and actions $h_t = (s_0, a_0, \ldots, s_{t-1}, a_{t-1}, s_t) \in \mathcal{H}$ is called a history and $\mathcal{H}$ is the set of all histories of any arbitrary length. In addition, after an action $a_t$ is executed, a reward $r_t = \rho(s_t, a_t) \in \mathbb{R}$ is observed.

A policy $\pi \in \Pi = \mathcal{S} \to \mathcal{P}(\mathcal{A})$ is a mapping from the state space $\mathcal{S}$ to the set of probability measures on the action space $\mathcal{P}(\mathcal{A})$, where $\pi(a|s)$ is the associated conditional probability density of action $a$ in state $s$. The function $J : \Pi \to \mathbb{R}$ is defined as the function mapping any policy $\pi$ to the expected

discounted sum of rewards gathered by an agent interacting in the MDP by sampling actions from the policy $\pi$. We call return of the policy $\pi$ the value provided by that function

$$J(\pi) = \frac{1}{1-\gamma} \mathop{\mathbb{E}}_{\substack{s \sim d^{\pi,\gamma}(\cdot) \\ a \sim \pi(\cdot|s)}} [\rho(s,a)] \ , \tag{1}$$

where $d^{\pi,\gamma}(\cdot)$ is the discounted state-visitation probability (Manne, 1960). In reinforcement learning, we seek to find an optimal policy $\pi^*$ maximizing the expected discounted sum of rewards $J$.

## 2.2 POLICY-GRADIENT ALGORITHMS

Policy-gradient algorithms (locally) optimize a parameterized policy $\pi_\theta$ to find the optimal parameter $\theta^*$ for which the return of the policy $J(\pi_{\theta^*})$ is maximized. Naively optimizing the parameterized policy by solely maximizing its return may provide sub-optimal results. This problem is mitigated in practice by implementing exploration strategies. These techniques consist in optimizing a surrogate learning objective $L$ that intrinsically encourages certain behaviors. In this work, we consider reward-shaping strategies where the expected discounted sum of rewards is extended by $K$ additional reward terms $\rho_i^{int}$, called intrinsic motivation terms, and optimize the learning objective

$$L(\theta) = \frac{1}{1-\gamma} \mathop{\mathbb{E}}_{\substack{s \sim d^{\pi_\theta,\gamma}(\cdot) \\ a \sim \pi_\theta(\cdot|s)}} \left[ \rho(s,a) + \sum_{i=0}^{K-1} \lambda_i \rho_i^{int}(s,a) \right] = J(\pi_\theta) + J^{int}(\pi_\theta) \ , \tag{2}$$

where $\lambda_i$ are non-negative weights for each intrinsic reward and where $J^{int}(\pi_\theta)$ is the intrinsic return of the policy. The parameter maximizing the learning objective is denoted by $\theta^\dagger$, which we distinguish from the optimal policy parameter $\theta^*$. Most of the intrinsic motivation terms can be classified in the two following groups.

**Uncertainty-based motivations.** It is common to provide bonuses for performing actions that reduce the uncertainty of the agent about its environment (Pathak et al., 2017; Burda et al., 2018; Zhang et al., 2021c). The intrinsic motivation terms are then proportional to the prediction errors of a model of the MDP dynamics. The latter model is usually learned.

**Entropy-based motivations.** It is also common to provide bonuses for visiting states and/or playing actions that are less likely in histories (Bellemare et al., 2016; Lee et al., 2019; Guo et al., 2021). In this work, we focus on two of these bonuses

$$\rho^s(s,a) = -\log d^{\pi_\theta,\gamma}(\phi(s)) \tag{3}$$

$$\rho^a(s,a) = -\log \pi_\theta(a|s) \ , \tag{4}$$

where $\phi(s)$ is a feature built from the state $s$. The corresponding intrinsic returns are maximized for policies that visit uniformly every feature, and for policies with uniformly distributed actions in each state, respectively. Note that these rewards require to estimate the distribution over the states and/or actions. Furthermore, they implicitly depend on the policy parameter $\theta$. The second technique is usually referred to as entropy regularization (Williams & Peng, 1991; Haarnoja et al., 2019).

In this work, we consider on-policy policy-gradient algorithms, which were among others reviewed by Duan et al. (2016) and Andrychowicz et al. (2020). These algorithms optimize differentiable parameterized policies with gradient-based local optimization. They iteratively approximate an ascent direction $\hat{d}$ relying on histories sampled from the policy in the MDP and update the parameters in the ascent direction, or in a combination of the previous ascent directions (Hinton et al., 2012; Kingma & Ba, 2014). For the sake of simplicity and without loss of generality, we consider that the ascent direction $\hat{d}$ is composed of the sum of an estimate of the gradient of the return $\hat{g} \approx \nabla_\theta J(\pi_\theta)$ and an estimate of the gradient of the intrinsic return $\hat{i} \approx \nabla_\theta J^{int}(\pi_\theta)$. In practice, the first is usually unbiased while the second is computed neglecting some partial derivatives of $\theta$ and is thus biased, typically neglecting the influence of the policy on the intrinsic reward.

## 3 STUDY OF THE LEARNING OBJECTIVE

In this section, we study the influence of the exploration terms on the learning objective defined in equation (2). We define two criteria under which the learning objective can be globally optimized

by ascent methods, and such that the solution is close to an optimal policy. We then graphically illustrate how exploration modifies the learning objective to remove local extrema.

## 3.1 POLICY-GRADIENT LEARNING OBJECTIVE

Policy-gradient algorithms using exploration maximize the learning objective function $L$, as defined in equation (2). We introduce two criteria related to this learning objective for studying the performance of the policy-gradient algorithm. First, we say that a learning objective $L$ is $\epsilon$-coherent when its global maximum is in an $\epsilon$-neighborhood of the return of an optimal policy. Second, we call learning objectives that have a unique maximum and no other stationary point pseudoconcave[1].

**Coherence criterion.** A learning objective $L$ is $\epsilon$-coherent if and only if

$$J(\pi_{\theta^*}) - J(\pi_{\theta^\dagger}) \leq \epsilon \, , \tag{5}$$

where $\theta^* \in \mathrm{argmax}_\theta J(\pi_\theta)$ and where $\theta^\dagger \in \mathrm{argmax}_\theta L(\theta)$.

**Pseudoconcavity criterion.** A learning objective $L$ is pseudoconcave if and only if

$$\exists! \, \theta^\dagger : \nabla L(\theta^\dagger) = 0 \wedge L(\theta^\dagger) = \max_\theta L(\theta) \, . \tag{6}$$

If the pseudoconcavity criterion is respected, there is a single optimum, and it is thus possible to globally optimize the learning objective function by (stochastic) gradient ascent (Bottou, 2010). If the learning objective is furthermore $\epsilon$-coherent, the latter solution is also a near-optimal policy, where $\epsilon$ is the bound on the suboptimality of its return.

## 3.2 ILLUSTRATION OF THE EFFECT OF EXPLORATION ON THE LEARNING OBJECTIVE

Exploration is of paramount importance when complex dynamics and reward functions are involved, where many locally optimal policies may exist (Lee et al., 2019; Liu & Abbeel, 2021; Zhang et al., 2021b). In the following, we first define an environment and a policy parameterization introduced by Bolland et al. (2023) that will serve as an example where it is possible to graphically illustrate the effect of exploration on the optimization process. For the sake of the analysis, we then represent the learning objectives associated with different exploration strategies, and depict their global and local optima. Learning objectives with a single global optimum respect the pseudoconcavity criterion. In addition, we represent the neighborhood $\Omega$ of the optimal policy parameters, such that any learning objective with its global maximum within this region is coherent for a given $\epsilon$. In light of the coherence and the pseudoconcavity criteria, we finally elaborate on the policy parameter computed by stochastic gradient ascent algorithms.

We consider the environment illustrated in Figure 1a where a car moves in a valley. We denote by $x$ and $v$ the position and speed of the car, both forming its state $s = (x, v)$. The valley contains two separate low points, positioned in $x_{initial} = -3$ and $x_{target} = 3$, separated by a peak. The car starts at rest $v_0 = 0$ at the highest low point $x_0 = x_{initial}$ and receives rewards proportional to the depth of the valley at its current position. The reward function is provided in Figure 1b. We consider a policy $\pi_{K,\sigma}(a|s) = \mathcal{N}(a|\mu_K(s), \sigma)$, namely a normally disturbed proportional controller with $\mu_K(s) = K \times (x - x_{target})$, parameterized by the vector $\theta = (K, \sigma)$. Figure 1c illustrates the contour map of the return of the policy as a function of the parameters $K$ and $\sigma$. The optimal parameters are represented by a black dot and correspond to a policy that drives the car so as to pass the peak and reach the lowest valley floor in $x_{target}$. The green area represents the set of parameters $\Omega = \{\theta' | \max_\theta J(\pi_\theta) - J(\pi_{\theta'}) \leq \epsilon\}$ for $\epsilon = 1$, and is used in the following discussion.

Figure 2 illustrates learning objectives combining the intrinsic rewards defined in equations (3) and (4) for different values of the weights $\lambda_1$ and $\lambda_2$. Here, the feature from equations (3) is composed of the position $\phi(s) = x$. First, we observe that for weights approaching zero, the parameter $\theta^\dagger$ maximizing the learning objective, represented by a black dot, corresponds to a policy with a high return. More precisely, it is in the green set $\Omega$ such that $\epsilon$-coherence is guaranteed for a small value of $\epsilon = 1$. Larger weights require larger values of $\epsilon$ for guaranteeing the $\epsilon$-coherence criterion. Nevertheless, when increasing the weights, we also observe that the learning objective eventually becomes pseudoconcave. There appears to be a trade-off between the two criteria. In Figure 2b,

---

[1]For the sake of simplicity, our definition slightly differs from that of Mangasarian (1975)

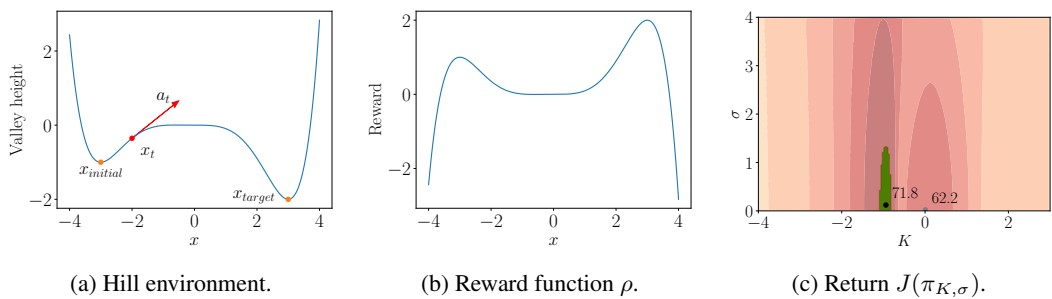

(a) Hill environment.  (b) Reward function $\rho$.  (c) Return $J(\pi_{K,\sigma})$.

Figure 1: Illustration of the *hill environment* in Figure 1a and its reward function in Figure 1b. In Figure 1c, the return of the policy $\pi_{K,\sigma}$ with the global and local maximum represented in black and grey, together with their respective return values.

we observe that in this environment, it is possible to find a learning objective that respects the pseudoconcavity criterion and the $\epsilon$-coherence criterion for $\epsilon = 1$. Indeed, there is a single global maximum in Figure 2b represented by a black dot that is furthermore part of the set $\Omega$.

Shaping the reward function with an exploration strategy based on the state-visitation entropy appears to be a good solution for optimizing the policy. However, a notable drawback is that the reward depends on the policy and its (gradient) computation requires to estimate a complex probability measure. In this example, the intrinsic reward function itself was estimated by Monte-Carlo sampling for every parameter, which would not scale for complex problems and requires approximations and costly evaluation strategies (Islam et al., 2019). In Appendix A, we present an alternative problem-dependent intrinsic reward, independent of the policy parameters and thus simple to compute efficiently, that still respects the pseudoconcavity and $\epsilon$-coherence criteria.

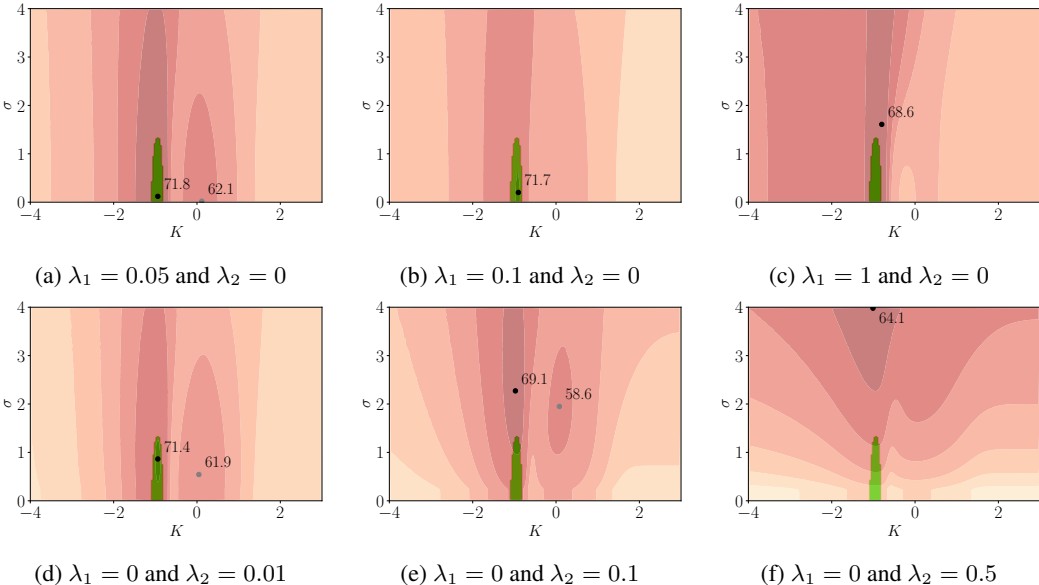

(a) $\lambda_1 = 0.05$ and $\lambda_2 = 0$  (b) $\lambda_1 = 0.1$ and $\lambda_2 = 0$  (c) $\lambda_1 = 1$ and $\lambda_2 = 0$

(d) $\lambda_1 = 0$ and $\lambda_2 = 0.01$  (e) $\lambda_1 = 0$ and $\lambda_2 = 0.1$  (f) $\lambda_1 = 0$ and $\lambda_2 = 0.5$

Figure 2: Contour map of (scaled) learning objective functions for different values of $\lambda_1$ and $\lambda_2$. The darker the map, the larger the learning objective value. The green area represents the set $\Omega = \{\theta' \,|\, \max_\theta J(\pi_\theta) - J(\pi_{\theta'}) \leq \epsilon = 1\}$, such that when the parameter maximizing the learning objective is part of $\Omega$, then the learning objective function is $\epsilon$-coherent with $\epsilon = 1$. The black dot is the parameter $\theta^\dagger$ globally maximizing the learning objective and the grey dot is the local (non-global) maximum of the learning objective if it exists. Both are labeled with the return values of the corresponding policies.

The observations suggest that well-chosen exploration strategies can lead to learning objective functions that satisfy the two criteria defined in the previous section, thereby guaranteeing that policies suboptimal by at most $\epsilon$ can be computed by local optimization. When designing exploration strategies, it is essential to keep in mind that we modify the learning objective for the algorithms to converge to optimal policy parameters, which can be achieved when both criteria are respected. While strategies such as enforcing entropy can be effective in some environments, they are only heuristic strategies and not to be relied upon exclusively. Furthermore, as illustrated, both criteria may be subject to a trade-off. In more complex environments, an efficient exploration strategy may require to balance both criteria, for example through a schedule on the learning objective weights.

## 4 STUDY OF THE ASCENT DIRECTION DISTRIBUTION

Optimizing pseudoconcave functions with stochastic ascent methods are guaranteed to converge (at a certain rate) under assumptions on the distribution of the gradient estimates at hand (Bottou, 2010; Chen & Luss, 2018; Ajalloeian & Stich, 2020). In this section, we study the influence of the exploration terms on this distribution in the context of policy gradients. More precisely, we study the probability of improving the learning objective, which, intuitively, shall be sufficiently large for the algorithm to be efficient. We formalize this intuition and illustrate how exploration strategies can increase this probability, leading to more efficient policy-gradient methods.

### 4.1 POLICY-GRADIENT ESTIMATED ASCENT DIRECTION

In general, gradient ascent algorithms update parameters in a direction $\hat{d}$ in order to locally improve an objective function $f$. The quality of these algorithms can therefore be studied (for a small step size $\alpha \to 0$) through the random variable representing the quantity by which the objective increases

$$X = f(\theta + \alpha\hat{d}) - f(\theta) = \alpha \langle \hat{d}, \nabla_\theta f(\theta) \rangle , \qquad (7)$$

where $\langle \cdot, \cdot \rangle$ is the Euclidean scalar product. This variable depends on the random event $\hat{d}$ estimated by Monte-Carlo simulations in practice. In order to study the convergence of gradient ascent algorithms, the expectation of $X$ is usually bounded by expressions involving the parameters of the algorithms. Doing so when gradients are biased is an active research field, where most results do not fit to our study. We therefore instead elaborate directly on the distributions $\mathbb{P}(\|X\|)$ that quantifies the magnitude of the variation of the objective function $f$, and $\mathbb{P}(X > 0)$ that quantifies when the ascent step improves this objective. In practice, the expectation of the random variable $X$ is positive and the estimate $\hat{d}$ is scaled and clipped by many algorithms, such that the sign of $X$ is arguably of more importance than its norm. In the following, we study $\mathbb{P}(X > 0)$ and assume it to be sufficient to measure the efficiency of optimization algorithms. In other words, we assume that all ascent steps lead to a constant variation of the objective, such that the rate of policy improvement is proportional to $\mathbb{P}(X > 0)$.

In the case of a policy gradient, we first assume that the return is pseudoconcave and that learning objectives respect the two previous criteria, and introduce two new criteria for the study of the policy improvement probability. The latter are independent (but not mutually exclusive) from those of Section 3. First, we say that an exploration strategy is efficient if following the ascent direction $\hat{d} \approx \nabla_\theta L(\theta)$ has a higher probability of increasing the return of the policy than following the direction $\hat{g} \approx \nabla_\theta J(\pi_\theta)$ for almost every $\theta$. Second, an exploration strategy is $\delta$-attractive if and only if, there exists a neighborhood of $\theta^\dagger$ containing the parameter $\theta^{int}$ maximizing the intrinsic return $J^{int}$, where the probability of increasing the return by following $\hat{d}$ is almost everywhere at least equal to $\delta$. Note that each probability measure and random variable is a function of $\theta$, which we do not explicitly write for the sake of keeping notations simple.

**Efficiency criterion.** An exploration strategy is efficient if and only if

$$\forall^\infty \theta : \mathbb{P}(D > 0) > \mathbb{P}(G > 0) , \qquad (8)$$

where $D = \langle \hat{d}, \nabla_\theta J(\pi_\theta) \rangle$ and $G = \langle \hat{g}, \nabla_\theta J(\pi_\theta) \rangle$.

**Attraction criterion.** An exploration strategy is $\delta$-attractive if and only if

$$\exists B(\theta^\dagger) : \theta^{int} \in B(\theta^\dagger) \wedge \forall^\infty \theta \in B(\theta^\dagger) : \mathbb{P}(D > 0) \geq \delta , \qquad (9)$$

where $\theta^{int} = \text{argmax}_\theta J^{int}(\pi_\theta)$, $B(\theta^\dagger)$ is a ball centered in $\theta^\dagger$, and $D = \langle \hat{d}, \nabla_\theta J(\pi_\theta) \rangle$.

First, the efficiency criterion quantifies if the exploration terms are collaborative with the original objective of maximizing the return. If this criterion is respected, estimating the gradient of the learning objective $\hat{d}$ rather than the return $\hat{g}$ will ensure a more likely policy improvement, which is desirable for efficient policy optimization. Second, the rationale behind the attraction criteria is that in many exploration strategies, the intrinsic reward is dense, and it is then presumably easy to optimize the intrinsic return in the sense that $\mathbb{P}(\langle \hat{i}, \nabla_\theta J^{int}(\pi_\theta) \rangle > 0)$. It implies that it is easy to locally improve the learning objective by (solely) increasing the value of the intrinsic motivation terms. It furthermore implies that policy-gradient algorithms may be subject to converging towards $\theta^{int}$ rather than $\theta^\dagger$ when $\mathbb{P}(\langle \hat{d}, \nabla_\theta J(\pi_\theta) \rangle > 0)$ is small. If the criterion is respected for large $\delta$, the latter is less likely to happen as policy gradients will eventually tend to improve the return of the policy if it approaches $\theta^{int}$ and enters the ball $B(\theta^\dagger)$; eventually converging towards $\theta^\dagger$.

## 4.2 ILLUSTRATION OF THE EFFECT OF EXPLORATION ON THE ESTIMATED ASCENT DIRECTION

Exploration is usually promoted and tested for problems where the reward function is sparse, typically in maze-environments (Islam et al., 2019; Liu & Abbeel, 2021; Guo et al., 2021). In this section, we first introduce a new maze-environment with sparse rewards where we illustrate the influence of exploration on the gradient estimates of the learning objective. To this end, we present two learning objective functions and report the likelihood that the gradient estimates improve the return as a function of the policy parameters. Based on these likelihood values, we elaborate on the influence of exploration on the performance of policy-gradient algorithms in the light of the efficiency and attraction criteria.

Let us consider a maze-environment consisting of a horizontal corridor composed of $S \in \mathbb{N}$ tiles. The state of the environment is the index of the tile $s \in \{1, \ldots, S\}$, and the actions consists in going left $a = -1$ or right $a = +1$. When an action is taken, the agent stays idle with probability $p = 0.7$, and moves with probability $1 - p = 0.3$ in the direction indicated by the action, then $s' = \min(S, \max(1, s + a))$. The agent starts in state $s = 1$ and the target state $s = S = 15$ is absorbing. Zero rewards are observed except when the agent reaches the target state where a reward $r = 1$ is observed. A discount factor of $\gamma = 0.99$ is considered. Finally, we study the policy going with probability $\theta$ to the right and probability $1 - \theta$ to the left, and $\forall s$ with density

$$\pi_\theta(a|s) = \begin{cases} \theta & \text{if } a = 1 \\ 1 - \theta & \text{if } a = -1 \end{cases}. \tag{10}$$

The return $J(\pi_\theta)$ is represented in black in Figure 3a as a function of $\theta$ along with two intrinsic returns, $J^a(\pi_\theta)$ in orange and $J^d(\pi_\theta)$ in green. The intrinsic reward $\rho^a(s, a) = -\log \pi_\theta(a|s)$ from equation (4) is used for computing $J^a(\pi_\theta)$, and the intrinsic reward $\rho^d(s, a) = (a - 1)/2$ is used instead for $J^d(\pi_\theta)$. The latter is a dense hand-crafted reward function penalizing actions taken from a suboptimal policy. In Figure 3b, we illustrate the return of the policy without exploration $J(\pi_\theta)$, along with two learning objective functions, $L^a(\theta)$ and $L^d(\theta)$, using as exploration strategies the intrinsic returns $J^a(\pi_\theta)$ and $J^d(\pi_\theta)$. We observe that the return is a pseudoconcave function with respect to $\theta$ and the optimal parameter is $\theta^* = 1$. In addition, the two learning objectives respect the $\epsilon$-coherence for $\epsilon = 0$, implying that $\theta^* = \theta^\dagger$, and respect the pseudoconcavity criteria. It is important to note with regard to the discussion from Section 3, there is no interest in optimizing the learning objectives rather than directly optimizing the return, as the latter is already pseudoconcave. In the following we illustrate how choosing a correct exploration strategy still deeply influences the policy-gradient algorithms when it comes to building gradient estimates.

Let us compute the estimate $\hat{g}$ and $\hat{d}$ relying on REINFORCE (Williams, 1992) by sampling 8 histories of length $T = 100$. We represent in Figure 4 the probabilities $\mathbb{P}(D > 0)$ for both learning objectives and $\mathbb{P}(G > 0)$. First, as can be seen for the learning objective $L^a(\theta)$, the probability of increasing the return by following the direction $\hat{d}$ is higher than the one of following $\hat{g}$ for small values of the parameter $\theta$. In this region of the parameter space, the efficiency criterion is respected. For the second learning objective $L^d(\theta)$ this criterion holds for any parameter. Second, concerning the attraction criterion, we represent at the top of Figure 4 the intervals $B^a = [\theta^{int,a}, \theta^{\dagger,a}]$ and $B^d = [\theta^{int,d}, \theta^{\dagger,d}]$. They correspond to the smallest balls containing the maximizers of the learning

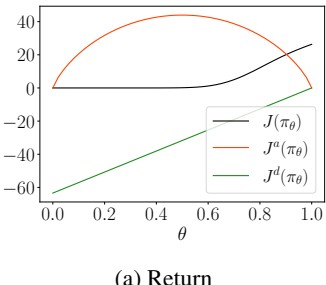

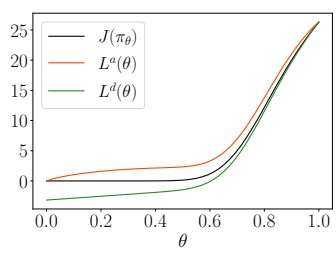

(a) Return                    (b) Learning objectives

Figure 3: Figure 3a represents the return of the policy along with two intrinsic return functions. In Figure 3b the return is also represented together with two learning objective functions, corresponding to the two intrinsic returns.

objective and of the intrinsic return. In addition, the minima of $\mathbb{P}(D > 0)$ over these intervals are also reported and denoted $\delta^a$ and $\delta^d$, for both learning objective respectively. By definition of the attraction criterion, it is thus respected for any values of $\delta$ at most equal to $\delta^a$ and $\delta^d$, for $L^a(\theta)$ and $L^d(\theta)$, respectively. All these observations can eventually be explained as the computation of $\hat{g}$ is always zero when the target is not reached, which is highly likely for policies with small values of $\theta$. Adding the exploration terms here leads to policy-gradient algorithms that compute more easily an optimal policy while naive optimization without exploration would fail or be sample inefficient.

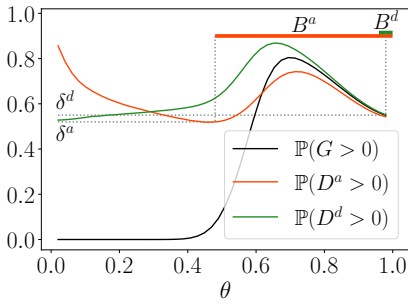

Figure 4: Illustration of the probability (estimated by Monte-Carlo) to improve the return by following the stochastic gradient estimates of $J(\pi_\theta)$, $L^a(\theta)$, and $L^d(\theta)$. At the top of the figure, the intervals $B^a = [\theta^{int,a}, \theta^{\dagger,a}]$ and $B^d = [\theta^{int,d}, \theta^{\dagger,d}]$ are represented. These intervals represent the smallest balls containing the parameters maximizing the learning objective and the intrinsic return, for both exploration strategies. Dotted lines represent the smallest probability over each interval to improve the return by following $\hat{d}$.

We have empirically shown that exploration in policy gradients does not only remove local extrema from the objective function, but also increases the probability that stochastic ascent steps improve the objective function. Under the previous assumptions, this probability measures the efficiency of algorithms. Furthermore, among different learning objectives respecting the coherence and pseudo-concavity criteria, it is best to choose one that has a high probability of being increased by stochastic gradient ascent. In Figure 4, the learning objective in green is better than the one in orange according to both criteria. Indeed, it is efficient over a larger set of policy parameters, and is $\delta$-attractive for a larger value of $\delta$. In the experiments, we used the naive REINFORCE estimates, yet the considerations generalize to any reinforcement learning technique where exploration can help to compute good estimates of the learning objective. Typically, estimating a critic by stochastic gradient ascent suffers from this problem as it is also built from an estimate computed from sampled rewards.

The problem discussed in this section strongly relates to a notion of overfitting or generalization in reinforcement learning. In situations where we sample with low probability some state and action pairs, the policy may be optimal over the set of pairs already sampled, the gradient estimates will then be zero with high probability, and the gradient updates will not lead to policy improvements. In the previous example, gradient estimates computed from policies with a small parameter value $\theta$

wrongly indicate that a stationary point has been reached as they equal zero with high probability. We quantify this effect with a novel definition of local optimality. We define as locally optimal policies over a space with probability $\Delta$ the policies that maximize the reward on expectation over a set of states and actions observed in a history with probability at least $\Delta$. Formally, a policy $\pi$ is locally optimal over a space with probability $\Delta$ if and only if

$$\exists\, \mathcal{E} \in \left\{ \mathcal{X} \,\Big|\, \int_{\mathcal{X}} d^{\pi,\gamma}(s)\pi(a|s)\, dads \geq \Delta \right\} : \pi \in \operatorname*{argmax}_{\pi'} \int_{\mathcal{E}} d^{\pi',\gamma}(s)\pi'(a|s)\rho(a,s)\, dads . \quad (11)$$

In the typical case of environments with sparse rewards, many policies observe with high probability state and action pairs with zero rewards and are thus locally optimal for large probabilities $\Delta$. Typically, in the previous example, the joint set $\{1,\dots,S-2\} \times \{-1,1\}$ is a set of state and action pairs $\mathcal{E}$ that respects the definition equation (11) for policies when $\theta$ is small for large values $\Delta$. Exploration mitigates the convergence of policy-gradient algorithms towards these locally optimal policies. Note that assuming a non-zero reward is uniformly distributed over the state and action space, exploration policies with uniform probabilities over visited states and actions are the best prior choice for sampling non-zero rewards with high probability. It can thus also be considered as the best choice of exploration to reduce the probability that the stochastic gradient ascent steps do not increase the objective value. Generally, such policy initialization priors may be learned from the framework developed by Lee et al. (2019).

## 5 CONCLUSION

In conclusion, this research takes a step towards dispelling misunderstandings about exploration through the study of its effects on the performance of policy-gradient algorithms. More particularly, we distinguished two effects exploration has on the optimization. First, it modifies the learning objective in order to remove local extrema. Second, it modifies the gradient estimates and increases the likelihood that the update steps lead to improved returns. These two phenomena were studied through four criteria that we introduced and illustrated.

These ideas apply to other direct policy optimization algorithms. Indeed, the four criteria do not assume any structure on the learning objective and can thus be straightforwardly applied to any objective function optimized by a direct policy search algorithm. In particular, for off-policy policy gradient, we may simply consider that the off-policy objective is itself a surrogate or that the gradients of the return are biased estimates based on past histories. Ideas introduced in this work also apply to other reinforcement learning techniques. Typically, for value-based RL with sparse-reward environments, convergence towards a value function outputting zero is expected with high probability. This is mostly due to the low probability of sampling non-zero rewards by Monte-Carlo. The discussions from Section 4 then apply, and a similar analysis can be performed.

Our framework opens the door for further theoretical analysis, and the potential development of new criteria. We believe that deriving practical conditions about the exploration strategies, and the scheduling of the intrinsic return, for guaranteeing fast convergence, should be the focus of attention. It could be achieved by bounding the policy improvement on expectation, which is nevertheless usually a hard task without strong assumptions. We furthermore believe that we provide a new lens on exploration necessary for interpreting standard exploration strategies, in the sense of designing surrogate learning objective, and developing new ones.

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

# A  ALTERNATIVE EXPLORATION STRATEGY

In this section, we provide additional remarks about exploration and provide an exploration strategy that guarantees coherence and pseudoconcavity of the learning objective from the environment in Section 3.

A notable drawback in the example about the environment from Section 3 is that the reward function depends on the policy parameters and its computation requires to estimate a complex probability measure. In Figure 5a, we illustrate an intrinsic reward bonus making the sum of rewards in equation (2) concave. The corresponding learning objective has a unique maximum, which is part of the set $\Omega = \{\theta' | \max_\theta J(\pi_\theta) - J(\pi_{\theta'}) \le \epsilon\}$ with $\epsilon = 1$. It can be seen in Figure 5b where the global maximum in black is within the set $\Omega$ in green. Both, the $\epsilon$-coherence and the pseudoconcavity criteria are thus respected for $\epsilon = 1$. Here, the reward $\rho^{int}$ is a simple function independent of the policy $\pi_\theta$. Finding such an intrinsic reward may be complex for other environments but the example underlines that exploration and reward shaping are mostly equivalent and that designing reward functions that are concave may help converging towards optimal policies.

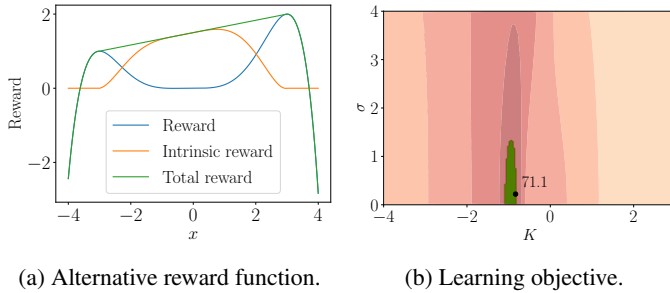

(a) Alternative reward function.          (b) Learning objective.

Figure 5: In Figure 5a, an alternative intrinsic reward function ensuring that the sum of rewards is a concave function. In Figure 5b, the contour function of the learning objective.

