# OpenReview forum: "Behind the Myth of Exploration in Policy Gradients"
_ICLR.cc/2024/Conference — Submitted to ICLR 2024_

### Official Review · Reviewer_yYKF · 2023-10-27

**Soundness:** 3 good
**Presentation:** 2 fair
**Contribution:** 2 fair
**Rating:** 5
**Confidence:** 3

**Summary:**

The authors use two pathological environments to explore aspects of exploration bonuses in objectives.  For example, they show that these bonuses can potentially eliminate local optima and make the loss landscape more amenable to SGD.

**Strengths:**

This work offers a compelling perspective and analysis of the topic.  While some of the ideas they explore might be "common knowledge" in the community, I believe that this might be the first time that some of these ideas have been explored thoroughly and scientifically.

**Weaknesses:**

Typos and minor suggestions:
- Discount factor backwards bracket in 2.1
- Is (1) missing the sum?
- “The valley is composed of two floors”: “floors” is a strange way to phrase it, and the intended meaning was initially unclear to me.  Consider rephrasing to something like "The valley contains two separate low points”.
- If there’s room, you could point to an appendix to briefly remind the reader why the middle and RHS of (7) are the same as alpha -> 0.  I know it’s fundamental SGD theory, but it confused me for a few minutes in this context (especially since, at first glance, the equation is not true, since it is just a first-order approximation and is only true as alpha approaches 0).
- In 4.2, the thorough description of the environment is nice, but I think it’s missing the time limit used.  Also, were the curves given in Figure 3 computed analytically or empirically?  Some more info might improve the paper.
In 4.2, you could remind the reader what (4) is so that they do not have to hunt for it.

Weaknesses:
1) Some of the questions below are clarity weaknesses or other possible weaknesses.
2) The contribution is entirely empirical (there are some nice theory concepts, but no proofs or useful properties are shown except empirically), and the empirical results are based entirely on extremely simple toy problems.  In my opinion, this does not invalidate their interesting perspectives on the issues that they illustrate with these results, but it does make the contribution less substantial.
3) The paper overall is a bit challenging to read.  For an example of a well-written bit that does *not* have this issue, the conclusion says “First, it modifies the learning objective in order to remove local extrema. Second, it modifies the gradient estimates and increases the likelihood that the update steps lead to optimal policies.”  This is great; more “high-level summary” passages like this in sections 3-4 would’ve made it much easier to read.  For an example of a hard-to-understand part, see the last question below.

Note: My primary concerns and slightly negative score come primarily from #2 (contribution), and the second question below ("reward-engineering").  The clarity weaknesses are not severe enough to have a large impact on my score.

**Questions:**

What is the meaning of the and symbol in (6)?  How is the gradient equal to “0 and L(theta)”?  Or is the and symbol meant to separate this line into two separate equalities?  If so, this is confusing, consider representing this in a more standard way, or using parentheses to disambiguate.  **Update:** upon reading more of the paper, and seeing this used more, I know the latter interpretation is correct, so no need to answer this question in your response, but I’ll leave this here to illustrate the potential confusion to the reader.

In 4.1, J^d seems less like an exploration term, and more like a “reward-engineering” term that simply makes the problem easier.  Am I missing a perspective on this?  This leads to a larger concern, in that much of the contribution of 4.1 hinges on this term, and I have doubts about whether this term can be legitimately thought of as an exploration bonus that is superior to the entropy bonus.

Can you please sum up the core take-away from 4.1?  I’ve reread the paragraph “We have empirically shown that exploration in policy gradients…” several times, but I’m struggling to understand exactly what I was supposed to take away from this section.  Is the point that the two criteria were good criteria in practice for choosing a good exploration bonus term?  If so, I am not convinced that this result will generalize beyond this specific toy setting (and the issue raised in the question above becomes even more of a concern in this case).  If not (or even if so), I think the intended take-away of this section needs to be spelled out more clearly.  **Update:** I understand better now upon a reread, so no need to address this question directly in your response.  However, the Section 4.1 paragraph noted is a perfect example of the last weakness noted above, so I’ll leave this question in the review.

---

> ### Author Response · Authors · 2023-11-21
> **Response Reviewer yYKF**
>
> Dear reviewer,
>
> Thank you for your valuable feedback, which we have carefully considered and integrated into the document.
> We've addressed most of the minor remarks and recognize the need for improved clarity on certain points.
> Notably, we've made substantial changes to Section 4.1 and the introduction, providing a more comprehensive overview of our contributions.
>
> We hope that these revisions have clarified the significance of our contributions to the RL community.
> With the manuscript modifications, we believe we now align with the acceptance standards of ICLR.
> Consequently, we kindly request you to reconsider your final decision.
>
> Below, we provide detailed responses to your specific questions.
>
> > The contribution is entirely empirical (there are some nice theory concepts, but no proofs or useful properties are shown except empirically), and the empirical results are based entirely on extremely simple toy problems. In my opinion, this does not invalidate their interesting perspectives on the issues that they illustrate with these results, but it does make the contribution less substantial.
>
> We acknowledge that the paper was rather vague on this point, but the criteria also introduce theoretical guarantees.
> Following your comment, Sections 3.1 and 4.1 have been reworked to better highlight the assumptions and results they imply.
> In particular, if our two criteria in Section 3 are met (note that we have changed the name of the second to better fit in the literature), it is possible to compute a policy by stochastic gradient ascent that is suboptimal by at most epsilon.
> Under the assumptions clarified in Section 4.1, the probability of improvement defines the probability of having a good solution after optimization.
> We agree, however, that there is future work to establish additional theoretical convergence guarantees and have introduced some avenues of reflection in conclusion.
>
> Regarding generalization, we acknowledge that our illustrations were conducted on relatively simple use cases.
> The low-dimensional spaces of states, actions, and parameters were chosen for clarity in visual representation.
> In addition, the experimental results were computed through exhaustive search in the parameter space combined with Monte-Carlo estimations, constraining the parameter space to low dimensions.
> Moreover, most exploration methods rely on function approximators for dealing with high dimensions, introducing complexities in learning dynamics that prevent isolating the sole effect of exploration.
> Nevertheless, we believe that the strongly non-linear dynamics in the first environment and the sparse reward function in the second reflects sufficient complexity from an optimization perspective.
>
> > The paper overall is a bit challenging to read. For an example of a well-written bit that does not have this issue, the conclusion says “First, it modifies the learning objective in order to remove local extrema. Second, it modifies the gradient estimates and increases the likelihood that the update steps lead to optimal policies.” This is great; more “high-level summary” passages like this in sections 3-4 would've made it much easier to read. For an example of a hard-to-understand part, see the last question below.
>
> We acknowledge your remarks concerning the clarity and have modified the paper substantially, including the two particular points you mentioned.
> In particular, the revised introduction aims to provide a clearer high-level view of the paper, related works, and contributions.
>
> > In 4.1, J^d seems less like an exploration term, and more like a “reward-engineering” term that simply makes the problem easier. Am I missing a perspective on this? This leads to a larger concern, in that much of the contribution of 4.1 hinges on this term, and I have doubts about whether this term can be legitimately thought of as an exploration bonus that is superior to the entropy bonus.
>
> Firstly, we have clarified in the introduction the distinction between exploration in policy gradient as intended in theory, i.e. optimizing a sufficiently stochastic policy, and how it is implemented in practice, i.e. involving reward shaping and optimizing a surrogate learning objective to increase stochasticity.
> In our paper, we analyse the second aspect by studying the exploration-based learning objective functions.
> Nevertheless, no assumption is made on the structure of the intrinsic rewards that may thus be built with any reward-shaping strategy.
> The analyses and discussions thus hold true for the expert-knowledge reward shaping in Section 4.1, which illustrates the potential achievements of exploration when well-designed for the problem at hand.
> We furthermore emphasize that all discussions are also validated on the standard entropy regularized objective.

---

### Official Review · Reviewer_qvTF · 2023-10-27

**Soundness:** 3 good
**Presentation:** 3 good
**Contribution:** 2 fair
**Rating:** 5
**Confidence:** 3

**Summary:**

The paper under review took a closer look at the value of exploration for RL algorithms from an analytic point of view.
On top of the intrinsic need to achieve global optimal (one has to know then entire environments),
authors proposed four criteria: __coherence, quasiconcavity, efficiency, and attraction__,
to measure the quality of an exploration, and demonstrated the effectiveness of these measures through simulations.

**Strengths:**

The idea of quantify effectiveness of an exploration is novel and well grounded.
The paper is well written, easy to follow. Simulations and plots are illustrative and helpful.

**Weaknesses:**

My concern lies in practicality. While intuitively, the proposed four criterion all make sense, however, for a general RL question, how to effectively compute these four measures is somehow challenging. For instance, following the paper's notation,  let $J$ be the objective function,
$J(\pi_{\theta^*})$ be the optimal, and $J(\pi_{\theta^\dagger})$ be the optimal with a way of exploration. Coherence requires
evaluating if $J(\pi_{\theta^*}) - J(\pi_{\theta^\dagger}) \leq \epsilon$.
Usually $\theta^*, \theta^\dagger$ are unknown, how would one utilize this coherence, and more generally the other proposed measures are not detailed in the paper.

**Questions:**

- Question on Practically: current simulations are good for readers to get the idea of the paper, but insufficient to demonstrate the capacity. It would be nice if the authors could take an concrete example, say a maze, with a few popular regularized objectives, compute (estimate) the proposed four criterion of these objectives, see which one is more effective based on these criterion. Then valid such prediction on the trained agent's behaviors.

---

> ### Author Response · Authors · 2023-11-21
> **Response Reviewer qvTF**
>
> Dear reviewer,
>
> We thank you for reviewing our paper and answer your two questions below.
> We believe these are due to a slightly unclear presentation that we have rectified in the revised paper version.
> Your remark gave us the opportunity to clarify this in the main paper, and we also better explain it below.
> We are confident that the paper offers a new way of analyzing exploration and is of interest to the ICLR community.
> We are receptive to any suggestions for improvements and kindly request a reconsideration of your decision for acceptance.
>
> > My concern lies in practicality. While intuitively, the proposed four criterion all make sense, however, for a general RL question, how to effectively compute these four measures is somehow challenging.
>
> The purpose of the paper is to provide a framework facilitating the interpretation of policy gradients and to identify relevant criteria for obtaining guarantees in policy gradients.
> As is often the case when theoretical criteria are proposed, their empirical verification is as complex as solving the problem at hand.
> If such criteria were easy to verify empirically, the problem would certainly be much more easy to solve too.
> However, we still believe that these criteria provide a good account of the effect of current exploration techniques, and their trade-offs.
> Moreover, we think that they provide some valuable, and often lacking, intuition for the design of new exploration techniques.
> We have added the development of more problem-specific bounds to our future work.
>
> > Current simulations are good for readers to get the idea of the paper, but insufficient to demonstrate the capacity. It would be nice if the authors could take an concrete example, say a maze, with a few popular regularized objectives, compute (estimate) the proposed four criterion of these objectives, see which one is more effective based on these criterion. Then valid such prediction on the trained agent's behaviors.
>
> The examples primarily serve to illustrate the criteria, which are derived from optimization theory.
> The experimental results were computed through exhaustive search in the parameter space combined with Monte-Carlo estimations, constraining the parameter space to low dimensions.
> Moreover, most exploration methods rely on function approximators for dealing with high dimensions, introducing complexities in learning dynamics that prevent isolating the sole effect of exploration.
> Nevertheless, we believe that the strongly non-linear dynamics in the first environment and the sparse reward function in the second reflects sufficient complexity from an optimization perspective.
> We have also clarified in the introduction that finding good exploration strategies is known to be problem specific and that we thus introduce a general framework for the study and interpretation of exploration in policy gradient methods instead of trying to find the best exploration method for a given task.

---

> ### Comment · Reviewer_qvTF · 2023-11-22
>
> Dear authors,
>
> Thank you for your response. It somewhat addressed my concern~
>
> I would like to acknowledge the contribution of theoretically defining the four criteria.
> But to have a practical impact, it feels some further work needs to be donem,
> either some further theoretically investigation (e.g. some sufficient condition to check these four criteria )
> or a more comprehensive simulation is needed to validate the proposed criteria.
>
> Otherwise, even we have these four criteria in mind, why these are good, how to design a policy that fit these etc are unclear.
> Hence, i decide to keep my original score.

---

### Official Review · Reviewer_kiQP · 2023-10-31

**Soundness:** 3 good
**Presentation:** 3 good
**Contribution:** 3 good
**Rating:** 6
**Confidence:** 3

**Summary:**

The paper studies the impact of exploration in policy optimization algorithms from an optimization perspective. The authors propose different measures of analysis, discuss implications of the techniques and measures proposed, and insights gathered on illustrative examples. Their conclusion is that exploration techniques that smoothify the objective (by adding entropy) eliminate local optima and increase the probability of trajectories through the optimization landscape reaching the goal.

**Strengths:**

I believe the properties of exploration bonuses on policy gradient algorithms are not entirely novel (for instance, their effect on smoothifying the optimization landscape was known, investigated empirically in \citet{ahmed19} and theoretically in \citet{mei2020}). The optimization perspective on exploration has been previously explored in \citet{mei2020, chung21, mei2022}. The main strength of the paper I believe lies in its clarity and rigor of collecting and expressing these scattered insights. I think the paper makes a good effort in systematically investigating these properties together, and offering a way of looking at them in the same place, on simple illustrative examples. Because of this, I think it has a lot of value to the community. Particularly, I believe the optimization view on exploration is still not known within the community, or understood by RL practitioners, so explaining these things with clarity is very relevant and subtly impactful. I also found the illustrative examples really interesting, particularly the last one.


@InProceedings{ahmed19,
  title = 	 {Understanding the Impact of Entropy on Policy Optimization},
  author =       {Ahmed, Zafarali and Le Roux, Nicolas and Norouzi, Mohammad and Schuurmans, Dale},
  booktitle = 	 {Proceedings of the 36th International Conference on Machine Learning},
  pages = 	 {151--160},
  year = 	 {2019},
  editor = 	 {Chaudhuri, Kamalika and Salakhutdinov, Ruslan},
  volume = 	 {97},
  series = 	 {Proceedings of Machine Learning Research},
  month = 	 {09--15 Jun},
  publisher =    {PMLR},
  pdf = 	 {http://proceedings.mlr.press/v97/ahmed19a/ahmed19a.pdf},
  url = 	 {https://proceedings.mlr.press/v97/ahmed19a.html},
  abstract = 	 {Entropy regularization is commonly used to improve policy optimization in reinforcement learning. It is believed to help with exploration by encouraging the selection of more stochastic policies. In this work, we analyze this claim using new visualizations of the optimization landscape based on randomly perturbing the loss function. We first show that even with access to the exact gradient, policy optimization is difficult due to the geometry of the objective function. We then qualitatively show that in some environments, a policy with higher entropy can make the optimization landscape smoother, thereby connecting local optima and enabling the use of larger learning rates. This paper presents new tools for understanding the optimization landscape, shows that policy entropy serves as a regularizer, and highlights the challenge of designing general-purpose policy optimization algorithms.}
}

@inproceedings{mei2020,
 author = {Mei, Jincheng and Xiao, Chenjun and Dai, Bo and Li, Lihong and Szepesvari, Csaba and Schuurmans, Dale},
 booktitle = {Advances in Neural Information Processing Systems},
 editor = {H. Larochelle and M. Ranzato and R. Hadsell and M.F. Balcan and H. Lin},
 pages = {21130--21140},
 publisher = {Curran Associates, Inc.},
 title = {Escaping the Gravitational Pull of Softmax},
 url = {https://proceedings.neurips.cc/paper_files/paper/2020/file/f1cf2a082126bf02de0b307778ce73a7-Paper.pdf},
 volume = {33},
 year = {2020}
}

@InProceedings{chung21,
  title = 	 {Beyond Variance Reduction: Understanding the True Impact of Baselines on Policy Optimization},
  author =       {Chung, Wesley and Thomas, Valentin and Machado, Marlos C. and Roux, Nicolas Le},
  booktitle = 	 {Proceedings of the 38th International Conference on Machine Learning},
  pages = 	 {1999--2009},
  year = 	 {2021},
  editor = 	 {Meila, Marina and Zhang, Tong},
  volume = 	 {139},
  series = 	 {Proceedings of Machine Learning Research},
  month = 	 {18--24 Jul},
  publisher =    {PMLR},
  pdf = 	 {http://proceedings.mlr.press/v139/chung21a/chung21a.pdf},
  url = 	 {https://proceedings.mlr.press/v139/chung21a.html},
  abstract = 	 {Bandit and reinforcement learning (RL) problems can often be framed as optimization problems where the goal is to maximize average performance while having access only to stochastic estimates of the true gradient. Traditionally, stochastic optimization theory predicts that learning dynamics are governed by the curvature of the loss function and the noise of the gradient estimates. In this paper we demonstrate that the standard view is too limited for bandit and RL problems. To allow our analysis to be interpreted in light of multi-step MDPs, we focus on techniques derived from stochastic optimization principles&nbsp;(e.g., natural policy gradient and EXP3) and we show that some standard assumptions from optimization theory are violated in these problems. We present theoretical results showing that, at least for bandit problems, curvature and noise are not sufficient to explain the learning dynamics and that seemingly innocuous choices like the baseline can determine whether an algorithm converges. These theoretical findings match our empirical evaluation, which we extend to multi-state MDPs.}
}

@article{Mei2021,
  author       = {Jincheng Mei and
                  Bo Dai and
                  Chenjun Xiao and
                  Csaba Szepesv{\'{a}}ri and
                  Dale Schuurmans},
  title        = {Understanding the Effect of Stochasticity in Policy Optimization},
  journal      = {CoRR},
  volume       = {abs/2110.15572},
  year         = {2021},
  url          = {https://arxiv.org/abs/2110.15572},
  eprinttype    = {arXiv},
  eprint       = {2110.15572},
  timestamp    = {Thu, 29 Jun 2023 16:58:03 +0200},
  biburl       = {https://dblp.org/rec/journals/corr/abs-2110-15572.bib},
  bibsource    = {dblp computer science bibliography, https://dblp.org}
}
@inproceedings{Mei2022,
 author = {Mei, Jincheng and Chung, Wesley and Thomas, Valentin and Dai, Bo and Szepesvari, Csaba and Schuurmans, Dale},
 booktitle = {Advances in Neural Information Processing Systems},
 editor = {S. Koyejo and S. Mohamed and A. Agarwal and D. Belgrave and K. Cho and A. Oh},
 pages = {17818--17830},
 publisher = {Curran Associates, Inc.},
 title = {The Role of Baselines in Policy Gradient Optimization},
 url = {https://proceedings.neurips.cc/paper_files/paper/2022/file/718d02a76d69686a36eccc8cde3e6a41-Paper-Conference.pdf},
 volume = {35},
 year = {2022}
}

**Weaknesses:**

I would have liked to see however how these methods generalize over problem instances, as it is known from \citet{mei2020} that PG methods are highly impacted by initialization and problem instances.
Some statements are rather vague, e.g. the authors claim that there is a trade-off between quasiconcavity and eps-coherence, but i’m not sure how general this statement is, and the authors do not provide proofs that guarantee these statements hold on all problem instances. At other times the authors use vague words like “appears” so its difficult to understand if these small illustrations would carry over to agents at scale, and are general enough.
The experiment illustrated in Fig 2 is interesting, but wasn’t this already known, the problem generally is that we cannot know the value of lambda beforehand as these are problem dependent, sometimes state-dependent and might also change over the optimization landscape for different policies on the way toward the optimal solution.


@inproceedings{mei2020,
 author = {Mei, Jincheng and Xiao, Chenjun and Dai, Bo and Li, Lihong and Szepesvari, Csaba and Schuurmans, Dale},
 booktitle = {Advances in Neural Information Processing Systems},
 editor = {H. Larochelle and M. Ranzato and R. Hadsell and M.F. Balcan and H. Lin},
 pages = {21130--21140},
 publisher = {Curran Associates, Inc.},
 title = {Escaping the Gravitational Pull of Softmax},
 url = {https://proceedings.neurips.cc/paper_files/paper/2020/file/f1cf2a082126bf02de0b307778ce73a7-Paper.pdf},
 volume = {33},
 year = {2020}
}

**Questions:**

The authors claim at some point that exploration strategies bias the policy. An exception is which we constrain the policy to stay in the vicinity of the previous policy, in which case we maintain optimality. It would have been interesting to explore such methods which are theoretically optimal.

---

> ### Author Response · Authors · 2023-11-21
> **Response Reviewer kiQP**
>
> Dear reviewer,
>
> We appreciate your feedback and the interest you give to our work.
> We have taken your comments into account and included your references into the document.
> We have significantly reworked the introduction and Section 4.1.
> Please find our detailed responses to your questions below.
>
> > I would have liked to see however how these methods generalize over problem instances, as it is known from \citet{mei2020} that PG methods are highly impacted by initialization and problem instances.
>
> Indeed, initialization plays a crucial role in the optimization.
> This influence is implicitly embedded in our criteria that consider the complete parameter space and in our examples for which all (potentially initial) parameters are analysed in the light of the criteria.
> Furthermore, the criteria can be extended and defined over a set of parameters where they hold true, such that policies initialized within this set would benefit from the properties and guarantees outlined in the paper.
> Section 3 illustrates that, with a fixed exploration strategy, gradient ascent can compute the optimal solution when we are in the "basin of attraction" of the global optimum.
> Additionally, the probabilities in the criteria from Section 4 may be low for certain parameters that will never be encountered by gradient ascent depending on the initialization, as illustrated in the example without exploration when the policy is initialized close to the optimum.
>
> Regarding generalization, we acknowledge that our illustrations were conducted on relatively simple use cases.
> The low-dimensional spaces of states, actions, and parameters were chosen for clarity in the visual representations.
> In addition, the experimental results were computed through exhaustive search in the parameter space combined with Monte-Carlo estimations, constraining the parameter space to low dimensions.
> Moreover, most exploration methods rely on function approximators for dealing with high dimensions, introducing complexities in learning dynamics that prevent isolating the sole effect of exploration.
> Nevertheless, we believe that the strongly non-linear dynamics in the first environment and the sparse reward function in the second reflects sufficient complexity from an optimization perspective.
>
> > Some statements are rather vague, e.g. the authors claim that there is a trade-off between quasiconcavity and eps-coherence, but i’m not sure how general this statement is, and the authors do not provide proofs that guarantee these statements hold on all problem instances. At other times the authors use vague words like “appears” so its difficult to understand if these small illustrations would carry over to agents at scale, and are general enough.
>
> As mentioned above, we have reworked the paper to clarify a number of points.
> Concerning the trade-off, in the general case, modifying the objective has no guarantee of preserving the global optimum.
> So, the greater the lambda weights, the greater the objective modification, and the greater the epsilon value.
> Similarly, if you modify the objective only slightly, the shape of the learning objective will not be greatly modified, and the _concavity_ remains unchanged.
> There is thus at best a trade-off, which we observe in the example, and at worst both criteria are violated.
>
> > The experiment illustrated in Fig 2 is interesting, but wasn’t this already known, the problem generally is that we cannot know the value of lambda beforehand as these are problem dependent, sometimes state-dependent and might also change over the optimization landscape for different policies on the way toward the optimal solution.
>
> Related works have been clarified in Section 1.
> A similar result had been stated (less formally) in the case of entropy regularization (maximum entropy RL) but had not been formalized nor stated in a general case to our knowledge.

---

> > ### Comment · Reviewer_kiQP · 2023-11-21
> > **Rebuttal response**
> >
> > Thank you for the response.
> >
> > Upon trying to reproduce the illustrations in the paper, I found that I could not. I do not believe there is sufficient information to reproduce them. You also did not provide any code for reproducibility, correct?
> >
> > Since the main contribution of the paper in these illustrations, without other theory or empirical results, the least the authors need to do is provide sufficient information in the paper, or a framework/codebase for others to test around hypotheses or exploration bonuses on the topic.
> >
> > From your results, I gather this avenue of reward bonuses is what you are exploring, so how can this be impactful to the community  in designing algorithms that better explore?

---

> > > ### Author Response · Authors · 2023-11-22
> > > **Details experiments**
> > >
> > > Please find bellow a detailed explanation about the procedure. We intend to provide access to a github repository that we will make as flexible as possible. If requested by the reviewer, we are also willing to add to the final version of the paper a short appendix with implementation details and highlight important hyperparameters.
> > >
> > > For the experiments in Section 3. We repeat for each parameter value in a grid the following operations. First, we sample policy trajectories in the environment given the parameter value. We then approximate the return with the mean of the discounted sum of rewards in a trajectory. The intrinsic return $J^a$ for the entropy regularization objective is computed by its definition. We compute the mean of the discounted sum of the log-likelihood of the actions in a trajectory, given the parameter value. The computation of the intrinsic return $J^s$ with the state-visitation entropy is slightly more intricate. Given the trajectories we compute a histogram of the visited states. Provided with this histogram, we compute for each state its (discretized) log-likelihood. The intrinsic return is afterwards computed as the average discounted sum of these log-likelihood estimations along trajectories. Zeros of the function and the Omega set are computed filtering the learning objective functions.
> > >
> > > For the experiments in Section 4. The same procedure is performed for sampling trajectories for each parameter of the policy. Given a trajectory, we compute the reinforce estimates of the gradient as follows. Knowing the actions are sampled from a Bernoulli distribution, we compute the score for each action analytically, and multiply the sum of scores by the sum of rewards for each trajectory. We average this estimate along eight trajectories to get the final gradient estimate. The latter step is probably the most important as it controls the variance of the estimate. Finally, plots report the ratio of the number of positive gradient estimates with respect to the number of estimates.

---

> > > > ### Comment · Reviewer_kiQP · 2023-11-23
> > > > **Re score**
> > > >
> > > > Dear authors,
> > > >
> > > > Thank you for your response.
> > > > I think this paper is insightful and a promising direction of investigation.
> > > > However, I tend agree with the other reviewers on requiring more evidence.
> > > >
> > > >
> > > > I will lower my original total score, which was a bit of an outlier.

---

### Meta-Review · Area_Chair_hvrB · 2023-12-07

**Metareview:**

The paper discusses exploration based on entropy bonuses in the context policy gradient methods. The focus in on two aspects: (i) the effect of exploration on the learning objective of a PG algorithm and (ii) distribution of gradient estimates.

Strengths:
- interesting, broadly applicable problem
- collects many existing insights about exploration in a single place

Weaknesses:
- paper is not currently completely reproducible (see review kiQP and discussion below). This is a crucial problem with an empirical-only work.
- toy experiments only
- no theory

Overall, I think the paper is promising, but not ready for ICLR at this time. In encourage the authors to: (i) improve reproducibility (ii) consider either adding more experiments or submitting to a venue with a slightly different bar for contributions (maybe TMLR?).

**Justification For Why Not Higher Score:**

Reproducibility is critical for this type of paper.

**Justification For Why Not Lower Score:**

N/A

---

### Decision · Program_Chairs · 2024-01-16

Reject